# Adaptation of *Bacillus subtilis* MreB Filaments to Osmotic Stress Depends on Influx of Potassium Ions

**DOI:** 10.3390/microorganisms12071309

**Published:** 2024-06-27

**Authors:** Simon Dersch, Peter L. Graumann

**Affiliations:** Centre for Synthetic Microbiology (SYNMIKRO), Fachbereich Chemie, Philipps-Universität Marburg, 35032 Marburg, Germany; simon.dersch@yahoo.de

**Keywords:** MreB, cell shape maintenance, *Bacillus subtilis*, osmotic stress, RodZ

## Abstract

The circumferential motion of MreB filaments plays a key role in cell shape maintenance in many bacteria. It has recently been shown that filament formation of MreB filaments in *Bacillus subtilis* is influenced by stress conditions. In response to osmotic upshift, MreB molecules were released from filaments, as seen by an increase in freely diffusive molecules, and the peptidoglycan synthesis pattern became less organized, concomitant with slowed-down cell extension. In this study, biotic and abiotic factors were analysed with respect to a possible function in the adaptation of MreB filaments to stress conditions. We show that parallel to MreB, its interactor RodZ becomes more diffusive following osmotic stress, but the remodeling of MreB filaments is not affected by a lack of RodZ. Conversely, mutant strains that prevent efficient potassium influx into cells following osmotic shock show a failure to disassemble MreB filaments, accompanied by less perturbed cell wall extension than is observed in wild type cells. Because potassium ions are known to negatively affect MreB polymerization in vitro, our data indicate that polymer disassembly is directly mediated by the physical consequences of the osmotic stress response. The lack of an early potassium influx response strongly decreases cell survival following stress application, suggesting that the disassembly of MreB filaments may ensure slowed-down cell wall extension to allow for efficient adaptation to new osmotic conditions.

## 1. Introduction

The shape of bacterial cells serves the purpose of allowing for the optimal adaptation to requirements within the niches of cells in their respective habitats [1]. Growth, as a coccus, allows for the generation of a densely packed 3D arrangement of cells, while directed motion requires a rod shape, and motion through viscous environments is strongly enhanced by curved/spiral architecture. Cell morphology in many bacteria depends on the activity of cell wall synthesizing enzymes, but also on a cytosolic, filament forming protein called MreB [2]. This actin-like protein affects cell wall synthesis via interaction with the synthesis machinery, and several models have been proposed as to how MreB filaments that move along roughly perpendicular paths underneath the cell membrane can affect the two dimensional and directional activity of cell wall synthesis proteins. MreB has been shown to form filamentous structure in vitro and in vivo. Many reports have established that the motion of MreB filaments correlates with the rate of cell wall growth [3,4,5], and thus, possibly, with the rate of cell wall extension. It would make sense that cell wall extension is regulated to follow the means of cells for metabolic power. Indeed, the supply of cell wall precursors plays an important role in the regulation of cell wall extension [6,7,8], and there are intriguing connections between MreB and central metabolic enzymes [9,10], or with a translation factor [11,12], or between the regulation of cell width and enzymes of the central carbon metabolism [13].

Interestingly, bacterial cells can survive and grow without a cell wall (e.g., many bacteria can form L-forms that propagate without peptidoglycan) [14], and if osmotically stabilized, protoplasts that were generated from bacterial cells by the degradation of their cell wall can survive and grow. It is therefore somewhat surprising that MreB is essential in rich medium [15]. Interestingly, in *Bacillus subtilis* cells, the essentiality of MreB can be overcome by increasing the magnesium concentration in the medium [16], an effect proposed to be based on the inhibition of autolysins that are inhibited by high magnesium concentrations, but in the absence of MreB would degrade the cell wall in an uncontrolled manner [17]. However, MreB has two paralogs, and it appears that the presence of at least one paralog is required for cell survival, even under high magnesium concentrations [18]. Thus, the question remains why MreB would be required for cells to survive if cellular architecture is highly important, but not crucial, for growth.

We have established that not only the speed of MreB filaments changes under different growth conditions, but that also the degree of assembly of MreB into filaments is affected by changes in the environment of cells, e.g., by changes in osmolarity [19]. The correlation between the filament assembly of MreB in response to osmotic stress has not yet been further analysed in terms of its molecular basis. It is possible that the proteins interacting with MreB mediate changes in filament assembly in response to osmotic shock, or possibly, filaments may be sensitive to changes in ion conditions arising during the adaptation to changes in osmolarity.

RodZ is a genuine MreB interactor, and strongly influences the assembly and dynamics of MreB filaments in *E. coli* cells in vivo. *RodZ* mutant *B. subtilis* cells show aberrant cell morphology, but not such a strong phenotype as mutant *E. coli* cells. RodZ is a membrane protein with a cytosolic domain that interacts with MreB, and may be a candidate for conveying changes in the adaptation to osmotic stress to MreB filaments. While MreB dynamics have been characterized in terms of the mobility of filaments and also at single molecule level, RodZ has not yet been studied at this molecular resolution.

In this work, we showed that RodZ protein also becomes less engaged with binding to its target proteins, i.e., MreB, upon osmotic upshift, in parallel with MreB filament disassembly. The latter also occurs in the absence of RodZ, indicating that stress-induced changes in MreB dynamics are not mediated via RodZ. On the other hand, the deletion of transporters that mediate the influx of potassium ions into cells as a first line of response to osmotic upshift abolished MreB filament disassembly, indicating a direct effect of intracellular potassium concentrations on filament formation and dissipation. We also investigated the effect of the mutations of MreB’s ATPase activity or filament contacts on stress response.

## 2. Results

### 2.1. MreB and RodZ Show Different Dynamics at Single Molecule Level That Are Differently Influenced by Osmotic Stress

MreB filaments have been shown to disassemble to a considerable extent following osmotic shock. As a measure of the formation of filaments, the motion of single MreB molecules can be analysed in cells by determining the percentage of molecules freely diffusing within cells, representing monomeric MreB, or of molecules diffusing at much reduced speed, representing filament-bound MreB; the proportion of filamentous (f-)MreB can be estimated under various environmental conditions. 

We wished to shed light into the molecular mechanism driving the changes in filament disassembly under osmotic stress conditions for MreB. Therefore, we studied the involvement of a known regulator of MreB filament dynamics, RodZ, as well as the importance of ions reported to affect MreB filament formation in vitro. MreB was visualized in a *B. subtilis* strain expressing a YFP-MreB fusion under the control of the inducible xylose promotor, using a low induction level of 0.01% xylose, from the ectopic amylase locus on the chromosome. We and others have shown that this strategy allows us to obtain insight into MreB dynamics without altering the cell’s physiology [3,4,20], which is easily done if the fusion can not complement for wild type MreB, or if MreB is overproduced [21].

Gaussian mixture modeling (GMM) was employed to analyse the tracks observed for YFP-MreB molecules. The probability density function (*y*-axis) of the occurrence of tracks showing certain displacements (*x*-axis) shows a non-Gaussian distribution (Figure 1A). Assuming the existence of two distinct populations, a low-mobility/slow-mobile population indicated by a dotted line, and a dynamic population indicated by the dashed line, can explain the overall observed distribution of tracks: the solid line indicated the two fits added together. The distribution of YFP-RodZ was visibly different from that of YFP-MreB (Figure 1B, see further below). The goodness of fit was close to “1” (Appendix A). The integral below the fits suggests that in exponentially growing cells, 60% of YFP-MreB molecules move with an average diffusion constant of 0.036 µm^2^ s^−1^, and 40% with D = 0.57 µm^2^ s^−1^ (Figure 1C). The latter is at the lower end of free diffusion observed for cytosolic proteins, and may be a mixture of freely diffusing cytosolic and membrane-attached MreB molecules. Indeed, MreB shows strong affinity to the cell membrane even in a non-polymerized form [22,23]. The slow-mobile population can be most easily explained with MreB being polymerized within filaments that have an average speed of rotation of 30 to 60 nm s^−1^ (note that single molecule tracking is carried out in the range of 20 ms per frame) [20,24,25,26]. 

The density function of the tracks became much wider after the addition of 1 M sorbitol (non-ionic osmotic stress) or of 0.5 M NaCl (ionic stress) to the growth medium (S7_50_ minimal medium) (Figure 1A), as described before. The slow-mobile fraction dropped from 60 to 43 or 44%, respectively (Figure 1C and Figure 2A), showing that almost half of the polymerized MreB fraction became diffusive.

We wished to investigate if changes in MreB polymerization might be accompanied by concomitant changes in proteins known to interact with MreB. We therefore analysed the single molecule dynamics of the integral membrane protein RodZ, an important interactor and possible membrane anchor of MreB [27,28]. A YFP-RodZ fusion has been shown to be able to complement the function of wild type RodZ [28,29]. We used the same strategy to visualize RodZ single molecule dynamics as with MreB: a YFP-RodZ fusion was expressed at very low levels under the control of the xylose promotor from the ectopic *amy* site. When utilizing a simultaneous GMM fit to identify diffusive populations, two populations were sufficient to describe the data, with R^2^ > 0.98 under all tested conditions (Figure 1B). Under normal growth conditions (S7_50_ media, OD ~0.6), the dynamics of the protein can be characterized as a slow-mobile fraction (D = 0.14 μm^2^s^−1^ ± 0.0012, fraction size: 47% ± 0.44) and a fast-mobile fraction (D = 0.61 μm^2^s^−1^ ± 0.0032, fraction size: 53% ± 0.44) (Figure 1D). These data suggest that, different from MreB, a (moderate) majority of RodZ molecules is unbound and thus diffusive, i.e., in a different equilibrium of bound versus unbound than MreB.

When inducing moderate osmotic stress, the population distribution shifted considerably, paralleling the shift seen for MreB: the mobile fraction increased to 66% (+0.5 M NaCl) or to 65% (+1 M sorbitol), with a concomitant decrease in the slow-mobile fraction (Figure 1B,D and Figure 2D). In contrast, two other MreB interactors, RodA and PbpH, did not show a significant shift in populations under stress [19]. 

This finding indicates two possible scenarios, (a) RodZ becomes less frequently bound to MreB filaments that strongly disassemble during osmotic stress (which we favour), or (b), MreB filaments disassemble because they lose contact to RodZ, which somehow responds to osmotic upshift conditions. 

STED imaging provides visual evidence for the disassembly of MreB filaments during stress, showing that only few discrete spots remain 30 min after the application of stress (Figure 2B). At this time point, cells became more kinked and curved, and the pattern of cell growth—visualized by HADA staining—became less homogeneous (Figure 2C), as was shown before [19]. HADA is a fluorescent-D-amino acid stain, which incorporates efficiently at sites of the de novo synthesis of peptidoglycan, to visualize the growth pattern via fluorescence microscopy.

For RodZ, we observed a shift from a more uniform localization within the cell membrane during exponential growth towards a patchy localization during stress adaptation (Figure 2E); note that cells continue to resume exponential growth about 2 h after the addition of osmotic compounds [19]. This pattern parallels that of the more patchy localization of MreB and of HADA staining during stress conditions (Figure 2B).

### 2.2. Loss of RodZ Does Not Influence MreB Dynamics Adaption during Osmotic Stress

Following these initial experiments, we set out to profile MreB dynamics during stress adaption in different backgrounds, to find out if other proteins may be required to convey changes to MreB dynamics and/or filament assembly. It has initially been reported that *rodZ* is an essential gene in *B. subtilis* [28], However, in recent years this view has changed [13,30]; in the absence of *rodZ*, cells have ang increased cell width and length. We used a *ΔrodZ* strain, generated by Boo et al. (2017) [31], and tracked YFP-MreB in this background (Figure 3A).

Through the loss of RodZ, cells became markedly more rounded, and larger in diameter (~1.3 μm, Figure 3D) compared to wildtype cells (average width ~0.8 µm), but the growth rate was not significantly impaired in S7_50_ minimal media, which is consistent with the observations made by other groups. We again employed a simultaneous two-population GMM fit to identify diffusive populations and relative shifts in distribution for different conditions (Figure 3A–C). As in wild type cells, MreB can be characterized as a slow-mobile or slow-mobile population with D = 0.038 μm^2^s^−1^ (63%) and as a faster, mobile population with D = 0.5 μm^2^s^−1^ (37%), with an R^2^ > 0.95 for all tested conditions (Appendix A). Thus, the diffusion constants and distribution of populations are in a similar range to those of MreB in a wild type cells. After inducing osmotic stress, the population distribution shifted towards the faster mobile population (+0.5 M NaCl: 56%, +1 M sorbitol 53%), similar to the changes observed for MreB in the wild type background. This shows that although RodZ showed a population distribution shift in response to osmotic stress comparable to that of MreB, its loss does not strongly interfere with the changes in single molecule dynamics seen for MreB in response to osmotic stress.

As judged from HADA staining, the pattern of PG-synthesis in *rodZ* mutant cells during normal growth conditions was similar to wild type cells (Figure 3D). Even though cells were larger in diameter when compared to the wildtype, PG-synthesis occurred relatively uniformly, with the weak continuous staining of the lateral cell wall and the stronger staining of division septa. When we induced osmotic stress by the addition of 0.5 M NaCl, the pattern of HADA staining changed to become more discontinuous, similar to the stress response observed in wild type cells. The demographs in Figure 2D show a summary of 100 cells, which can be compared to the demographs in Figure 1. In conclusion, the loss of RodZ did not significantly influence changes in MreB dynamics in response to osmotic stress, nor the general PG-synthesis patterns under normal growth.

### 2.3. Disassembly of MreB Filaments during Adaptation to Osmotic Depends on Potassium Uptake

Ionic conditions can have a large impact on the formation of MreB filaments in vitro [23]. While divalent ions strongly support filament formation, monovalent kations have an opposing effect. As a first response to a rapid increase in extracellular ion/molecule concentration, *B. subtilis* cells rapidly import potassium ions [32]. Later, potassium is again exported and replaced by compatible solutes made by cells or also taken up from the surroundings. We hypothesized that a first influx of potassium may be fully or partially responsible for the disintegration of a large fraction of MreB filaments.

KimA is a moderate-affinity K^+^/H^+^ symporter involved in potassium homeostasis [33]. To test if and how MreB dynamics adapt to osmotic stress in a *kimA* null mutant, we moved the YFP-MreB fusion into the corresponding mutant background. GMM evaluation suggested the existence of two populations (Figure 4A), one showing high mobility with D = 0.35 μm^2^s^−1^, and a slower one with D = 0.046 μm^2^s^−1^. Note that diffusion constants are different because simultaneous fits are applied to the different conditions, such that changes in dynamics are reflected by changes in population sizes only, and not also in changes in diffusion constants—this evaluation is more suitable to compare the dynamics of the same molecule in different conditions. In unstressed cells the mobile fraction had a size of ~48%, which changed to around 59% after the addition of 0.5 mM NaCl, and to about 62% after incubation with 1 M sorbitol (Figure 4A–C). These changes are comparable to MreB dynamics and stress adaption in the WT background. Similarly, HADA staining and the subsequent evaluation of the resulting demographs revealed a pattern that was closely matching to PG-synthesis in the WT background, with the de novo cell wall synthesis becoming disorganized during osmotic stress.

KtrAB, a high affinity monovalent cation transporter and KtrCD a lower affinity homolog, are highly important for K^+^ homeostasis and are known to be involved in rapid K^+^ uptake during osmotic stress [34,35]. KtrA and C are the peripheral membrane-associated subunits, which regulate the activity of KtrB and D, the transmembrane channel-forming subunits, respectively. When both *ktrAB* and *ktrCD* are disrupted, cells become highly susceptible to osmotic stress. Even cells lacking only the high affinity KtrAB cannot fully recover from an osmotic shock of over 0.6 M NaCl, while wild type cells are able to do so rather effortlessly [35].

Strikingly, the single-molecule dynamics of YFP-MreB did not show strong changes during stress adaptation in a *ΔktrB* background. MreB displayed a slow-mobile population of 53% and a mobile population of 47%. However, during osmotic stress no considerable shift in population distributions was observed, with 45% mobile (NaCl) or 46% (sorbitol) (Figure 5A–C). Intriguingly, the pattern of PG-synthesis, as observed via HADA staining, also appeared visibly less disrupted and more homogeneous than in wild type cells (compare demograph of Figure 2C and Figure 5D). Thus, these findings suggest an influence of rapid K^+^ ion influx on MreB dynamics and population distribution, and a change in the pattern of cell wall synthesis in vivo.

In order to shed light on the question whether this effect may be specific for the KtrAB system, or more broadly dependent on potassium influx, we also monitored MreB dynamics in a *ΔktrD* background (Appendix A). Similar to *ΔktrB* cells, *ΔktrD* cells did not show changes in MreB single molecule motion in response to osmotic stress. Closely matching the data obtained for *ΔktrB* cells (Figure 5A–C), population sizes of mobile and slow-mobile MreB molecules did not change in response to NaCl or sorbitol stress (Appendix A). Also, changes in HADA staining were less pronounced than in wild type cells (Appendix A). Thus, MreB did not show disassembly kinetics in cells lacking a strong influx of potassium ions, clearly showing that intracellular potassium concentrations have a profound influence on MreB assembly. In parallel, cells lose the ability to efficiently adapt to changes in osmolarity. We propose that the disassembly of MreB filaments may slow down cell wall extension and thereby help cells to first cope with adaptation to new osmolarity before growth is resumed.

### 2.4. MreB Mutants Show Aberrant Adaption to Osmotic Stress

MreB A277Q/K305D and MreB D158A are two notable MreB mutant forms that disturb filament formation and have a dominant negative effect on wild type filaments [20,36]. The combined point mutations MreB A277Q/K305D in the known subunit contact site of the protein leads to a reduced filament size, the more spotty localization pattern of MreB and subsequently helical, or increasingly bendy, cells [20]. MreB D158A is a mutation in the ATP-binding pocket of the protein, which leads the formation of a smaller number of observable filaments, which are wider and less dynamic than wild type filaments. We wanted to investigate how impaired MreB filaments would alter the response to moderate osmotic stress conditions. To this end, we tracked YFP fusions of both mutants, expressed from the ectopic *amy*-locus under low expression levels, as was carried out above with wild type MreB. Likewise, a two-population fit was sufficient to describe the data reliably (R^2^ > 0.98).

Both mutant MreBs showed much smaller fraction sizes for slow-mobile molecules compared to the wildtype (MreB A277Q/K305D: D = 0.046 μm^2^s^−1^ at 39%; MreB D158A: D = 0.049 μm^2^s^−1^ at only 11%) (Figure 6A, Figure 7A and Appendix A), versus 69% for wild type MreB (Figure 2A). Thus, most of the molecules were present in the mobile population, i.e., freely diffusive, especially in the case of MreB D158A, which was only present as fMreB to 11%. This is in agreement with the smaller amount of filaments observed via STED microscopy (Figure 6B and Figure 7B), and with the epifluorescence microscopy reported for both mutants [20]. The induction of osmotic stress with 1 M sorbitol caused a small shift towards the fast-mobile population for both mutants, to 64% for MreB A277Q/K305D and to 94% ± 0.024 for MreB D158A, indicating further filament disassembly. However, when osmotic stress was induced with NaCl, however, mutants showed a strong increase in their slow-mobile populations: from 39 to 58% for MreB A277Q/K305D (Figure 7A) and from 11 to 58% for MreB D158A. Thus, in both mutant forms, in which efficient filament formation is impaired, the reaction to osmotic stress was strikingly similar, and dissimilar to wild type MreB. These experiments show that the disassembly of MreB filaments via potassium influx depends on proper kinetics for filament formation.

## 3. Discussion

There are many molecular pathways for conveying the disturbance of homeostasis to response modules in all kinds of cells. In bacteria, kinases can sense perturbations or alterations in environmental conditions and active response regulators to trigger appropriate responses, e.g., in transcription regulation. Besides a myriad of molecular switches, many if not most of which are based on nucleotide hydrolysis, cells can also employ biophysical changes that directly lead to alterations in biological processes. The temperature-dependent unfolding of mRNA generates RNA thermometers that can trigger, between a few degrees Celsius [37], mechanosensitive proteins are involved in cellular responses within our immune system [38]. In this work, we show that bacterial MreB may also represent a sensor of biophysical properties, and its sensitivity towards potassium ions may be harvested to respond to osmotic shock conditions. 

MreB has been shown to play a central role in the control of rod-shaped growth in many bacterial species [39]. Much is known on its connection with the cell wall synthesis machinery, and with enzymes providing the precursors for cell wall extension, but the exact way in which the rotation of MreB filaments affects the rate of cell wall synthesis is still unclear. It has been shown that the directional motion of MreB filaments and the reversal of the direction correlates with the motion of PG synthesizing enzymes, and that filament dynamics correlate with growth speed [19,24,40]. Together with many previous observations, these data suggest that in *B. subtilis*, MreB filaments may play a crucial role in directing the speed of cell wall elongation and in coordinating all the enzymes involved in PG extension during unperturbed growth.

We have recently begun to uncover that MreB may have additional roles in the physiology of bacterial cells that are important when growth conditions become suboptimal for cell proliferation. We have found that osmotic upshift leads to the disassembly of about 50% of fMreB, which is accompanied by a transient loss of the regular cell shape during the adaptation phase [19]. A total of 2 h after the application of osmotic stress, cell wall synthesis regained its regular pattern and MreB filaments returned to their normal appearance. We propose that the transient loss of MreB filament formation slows down cell wall growth, allowing cells to adopt to new osmotic conditions. While this hypothesis still needs proof, we have addressed the question why MreB filaments might disassemble during osmotic stress conditions. We tested two hypotheses: (a) MreB filaments disassemble because a major interactor and modulator of MreB filaments, RodZ, might respond to osmotic upshift conditions, or (b) MreB filaments might be affected by changes in osmolarity per se. In this respect, there is an intriguing parallel: as a first response to an increase in extracellular osmolarity, *B. subtilis* cells allow potassium ions to enter the cytosol as a first means to counterbalance the influx of water into cells [41]. Because potassium ions per se are problematic for cells, this first wave of influx is reversed and compensated for by the production and import of compatible solutes, such as glycine betaine or proline or trehalose [32]. Interestingly, the assembly of MreB filaments is negatively affected by monovalent kations, and thus greatly by potassium [23]. To test hypothesis A), we investigated if MreB filaments disassemble in the absence of RodZ in response to osmotic upshift, which was the case, ruling out a direct role of RodZ in this process. Interestingly, RodZ motion was strongly affected by osmotic stress: the protein shifted from 47% molecules being bound to a larger complex (i.e., MreB filaments) to 34%, showing that, like MreB, RodZ became more freely diffusive. It is very likely that RodZ becomes more diffusive because MreB filaments disintegrate during osmotic upshift, leading to the loss of a major interaction target for RodZ.

To test if potassium influx might affect fMreB, we tested deletions of two major potassium importers, KtrAB and KtrCD, that play a key role in the osmotic stress response in *B. subtilis* [32]. We observed no effect on single molecule dynamics of MreB, i.e., no disassembly occurred in cells lacking strong potassium influx. Because MreB filament disassembly was seen when a minor potassium channel (KimA) was deleted, our data show that strong potassium influx is triggering filament disintegration. Thus, changes in MreB single molecule dynamics depend on potassium influx, which in turn is essential for adaptation to increased osmolarity: *ktrAB* or *ktrCD* deletion strains show strong defects in growth after osmotic upshift [35]. This could be due to an increase in protein aggregation because of water loss following osmotic upshift. However, we propose that MreB could play an active role in stress adaptation: we observed that the pattern of cell wall synthesis becomes spotty as opposed to relatively uniform along the lateral cell wall, which is not the case (or less pronounced) in *ktrB* or *ktrD* mutant cells. This indicates that during osmotic adaptation, cell wall synthesis becomes less uniform and thus possibly less effective, allowing cells to adapt to new osmotic conditions in the environment by slowed-down synthesis. Indeed, cell wall synthesis enzymes have been shown to be differently active at different pH conditions [42,43], or, for example, inhibited at high magnesium concentrations (autolysins) [17], such that the cell wall synthesis machinery has the capacity to operate at different environmental conditions [43], However, with the need to adapt after a shift from optimal growth conditions. 

We also investigated if mutations in MreB that affect ATPase activity, or lead to the weakening of subunit contacts, change the dynamics of MreB at a single molecule level. Reduced ATPase activity led to a severe loss of filament formation, as did—to a lesser but considerable extent—the interface mutations. However, interestingly, both mutant forms showed aberrant changes to osmotic upshift: the addition of 1M sorbitol as non-ionic osmotic stress further disassembled MreB D158A (ATPase mutant) filaments, or had no effect on MreB A277Q/K305D (interface mutant), while sodium chloride upshift led to a strong increase in filament formation for both mutant versions. Thus, the effect of potassium influx on MreB filaments depends on proper filament architecture, i.e., the biochemical properties of fMreB. These findings further support the idea that MreB filaments show a specific response to changes in intracellular ion concentrations, helping cells to adapt to new stress conditions. The latter hypothesis is difficult to test: *mreB* mutant cells grow poorly, even in the presence of increased magnesium concentrations, which may be based on the pleiotrophic effects caused by the absence of MreB.

Thus, our work uncovers that the changes in MreB filaments occurring to osmotic upshift depend on potassium influx, are affected by mutations changing filament architecture, and are paralleled by changes in the pattern of PG synthesis. These observations extend the repertoire of MreB filament dynamics and point to a possible role of MreB during adaptations to changes in the environment, in addition to a putative scaffolding role during cell wall extension under non-perturbed growth conditions.

## 4. Methods

### 4.1. Growth Conditions

Cells were inoculated from overnight culture in S7_50_ minimal media (fructose as carbon source) with the respectively appropriate antibiotics (100 mg/mL ampicillin, 100 mg/mL spectinomycin, 5 mg/mL chloramphenicol) at 30 °C shaking (200 rpm), and grown to the exponential phase (OD ~0.6). The expression of genes under the control of the p*xyl*-promotor was induced by the addition of xylose to a final concentration of 0.01%. To induce osmotic stress, NaCl or sorbitol were added to a final concentration of 0.5 M or 1 M, respectively, and, if not stated otherwise, cells were imaged 30 min after induction. If not stated otherwise, 4 µL of exponentially growing cell culture (OD ~0.6) were spotted on glass coverslips (Roth) and fixed with a S7_50_ agarose pad (1% *v*/*w*), mounted on the microscope and subsequently imaged. 

### 4.2. Construction of Strains

The YFP-MreB fusion (expressed from the ectopic *amyE* site, using very low xylose induction to express minute amounts of YFP-MreB in addition to the 2000 to 3000 MreB molecules per cell) was taken from [20], and likewise both mutant *mreB* alleles. The YFP-RodZ construct was generated by a single crossover integration of pHJDS (Defeu Soufo and Graumann, 2006) into which the first 500 bp of *rodZ* had been cloned using *Eco*RI/*Apa*I restriction sites. The generated plasmid was integrated into the *B. subtilis* chromosome by single crossover integration. This generated a construct, in which the transcription of *yfp-rodZ* is driven by the *pxyl* promoter, and the original promoter drives the expression of an N-terminally truncated *rodZ* gene. The integrity of the strain was verified by (a) the sequencing of a PCR fragment generated by primers binding within *pxyl* (driving the expression of *yfp-rodZ*) downstream of the 500 inserted *rodZ* bases (ensuring proper genome integration), and by Western blotting, showing a band at 52 kDa (ensuring that full length YFP-RodZ is produced). Mutant strains (*rodZ*, *ktrB*, *ktrD*, *ΔkimA*) were generated by transforming respective deletion strains (*B. subtilis* mutant collection) with chromosomal DNA from the strain expressing YFP-MreB from the *amyE* site (see above). 

### 4.3. Microscopy and Image Analysis

Slim field microscopy was performed on a customized Nikon Eclipse Ti microscope setup, (100x oil-immersion objective, NA = 1.49) where the laser was aligned to the back focal plane of the object. Fluorescent protein fusions were illuminated using a 514 nm laser diode beam line of 100 mW power (160 W cm^−2^ on image plane) to first bleach most of the fluorophores, and subsequently track single molecules (events with a single bleaching step). A high-refresh EMCCD camera (ImagEM X2 EM-CCD, Hamamatsu) and frame transfer mode was used to acquire 20 ms stream acquisitions over 30 s per movie. Epifluorescence microscopy was performed using a Zeiss Axio Observer Z1 (100x oil-immersion, 1.46 NA, Photometrics Cascade II:512 camera). Gated-STED microscopy was performed using a Leica SP8 confocal microscope with a CW laser (200 hz, 4-line averages, unidirectional scanning). 

For this study we chose the Gaussian mixture model (GMM) as the basis for analysis, which was also used in our initial publication on the motion of MreB and cell wall synthesis enzymes. GMM can explain a given set of single molecule displacement data by fitting the most likely superposition of, in our case, up to three Gaussian curves, as molecules might exhibit multiple diffusive sub-populations, depending on their state (for example DNA-bound state vs. freely diffusive state). Using GMM we can infer the distinct fraction sizes and diffusion constants of the respective populations of a given molecule from the probability density of displacements. 

SMTracker allows for a simultaneous fit, where the subpopulation diffusion constants *Dx* over different states are identical, and changes are reflected in the variation in fraction sizes *αx*. In this case, the simultaneously set subpopulation diffusion constants *Dx* are determined by the best fit over all compared conditions. This allows us to directly compare the changes in a molecule’s movement profile over different treatments or conditions via percentile changes in the distinct sub-populations fraction sizes.

The goodness of the Gaussian fits was determined by comparing the observed with the predicted data spreads, and for all tested conditions the R^2^ was at least 0.98, indicating that the data quality was sufficient.

Demographs were generated with Oufti [44] from *n* = 100 cells each. Demographs of *n* = 100 cells, where each cell is represented by a one pixel wide line of its integrated fluorescence signal. The cells are identified by polygonal cell meshes in Oufti, sorted by length and segmented. The fluorescence signal for each corresponding segment is normalized by the fluorescence of the brightest segment of each cell. The segments are then plotted as a one-pixel wide heat map from 0 (no signal) to 1 (maximum-signal).

### 4.4. Fluorescent D-Amino Acid (FDAA) Labeling

FDAA-labeling to visualize cell wall synthesis was performed utilizing HADA, according to [45]. HADA was added to a final concentration of 0.5 mM, followed by incubation for 20 min, shaking (30 °C, 200 rpm, tubes were wrapped in aluminum foil to block out light), and subsequent washing three times with PBS, immediately before mounting and imaging.

## Figures and Tables

**Figure 1 microorganisms-12-01309-f001:**
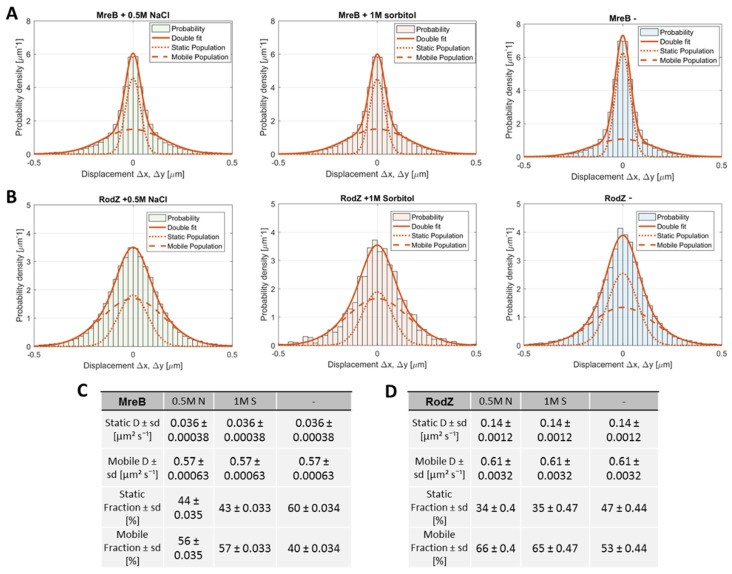
Two-population Gaussian mixture model (GMM) fit of displacement versus probability density for MreB (**A**) and RodZ (**B**) and calculated diffusion values and fraction sizes (**C**,**D**) for unstressed cells (-), and cell growth with added 0.5 M NaCl or 1 M sorbitol.

**Figure 2 microorganisms-12-01309-f002:**
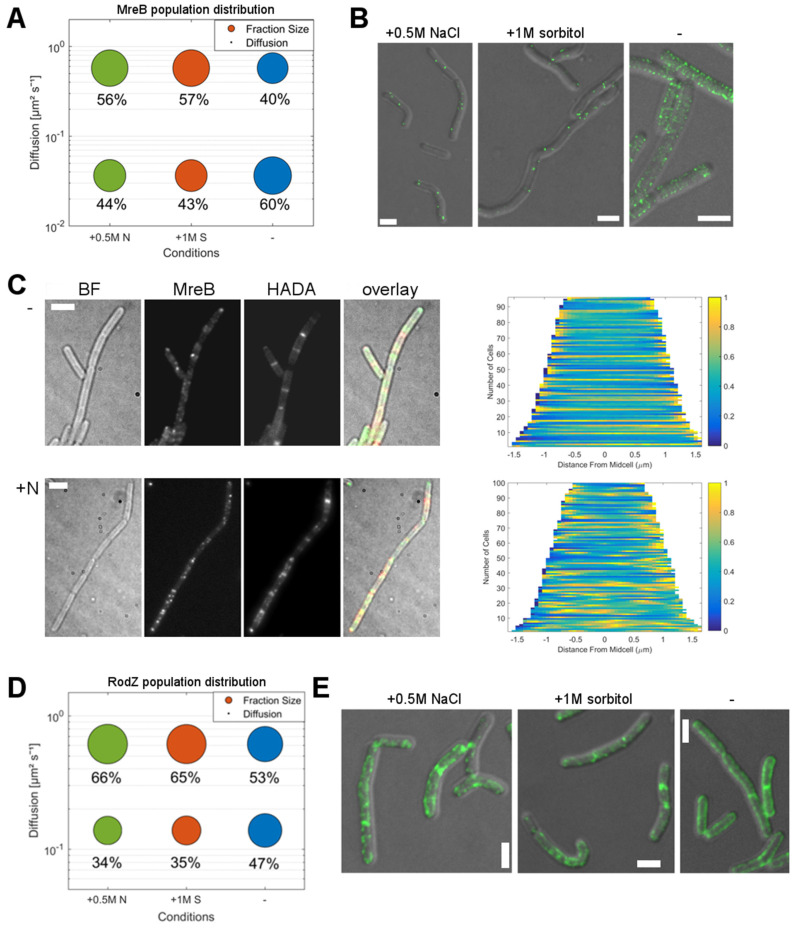
Adaption of MreB and RodZ dynamics in response to osmotic stress. (**A**): Bubble plot of YFP-MreB diffusion and population distribution based on a two-population Gaussian mixture model (GMM) fit of displacement vs. probability density under normal growth conditions (blue circles) (-) and with the addition of 500 mM NaCl (green circles) (+0.5 M N) or 1 M sorbitol (red circles) (+1 M S); (**B**): STED images of *B. subtilis* PY79 cells expressing YFP-MreB under the control of the xylose promotor (+0.01% xylose) in S7_50_ minimal media, under normal growth conditions (-) and with the addition of 500 mM NaCl (+N) or 1 M sorbitol (+S), scale bar 2 μm; (**C**): Bright field (BF), YFP-MreB (green-channel), HADA (red-channel, 0.5 mM) and overlay images of *B. subtilis* cells in the exponential phase, expressing YFP-MreB under the control of the xylose promotor (+0.01% xylose) in S7_50_ media and the corresponding demograph of the distribution of HADA signal throughout *n* = 100 cells (20 min HADA staining without stress (-) and with added 0.5 M NaCl (“N”), images taken after washing 3 times with PBS), scale bar 2 μm; (**D**): Bubble plot of YFP-RodZ diffusion and population distribution, based on a two-population Gaussian mixture model (GMM) fit of displacement vs. probability density under normal growth conditions (-) and with the addition of 500 mM NaCl (+0.5 M NaCl) or 1 M sorbitol (+1 M S); (**E**): STED images of *B. subtilis* PY79 cells expressing YFP-RodZ under the control of the xylose promotor in S7_50_ minimal media, under normal growth conditions (-) or with the addition of 500 mM NaCl or 1 M sorbitol, scale bar 2 μm.

**Figure 3 microorganisms-12-01309-f003:**
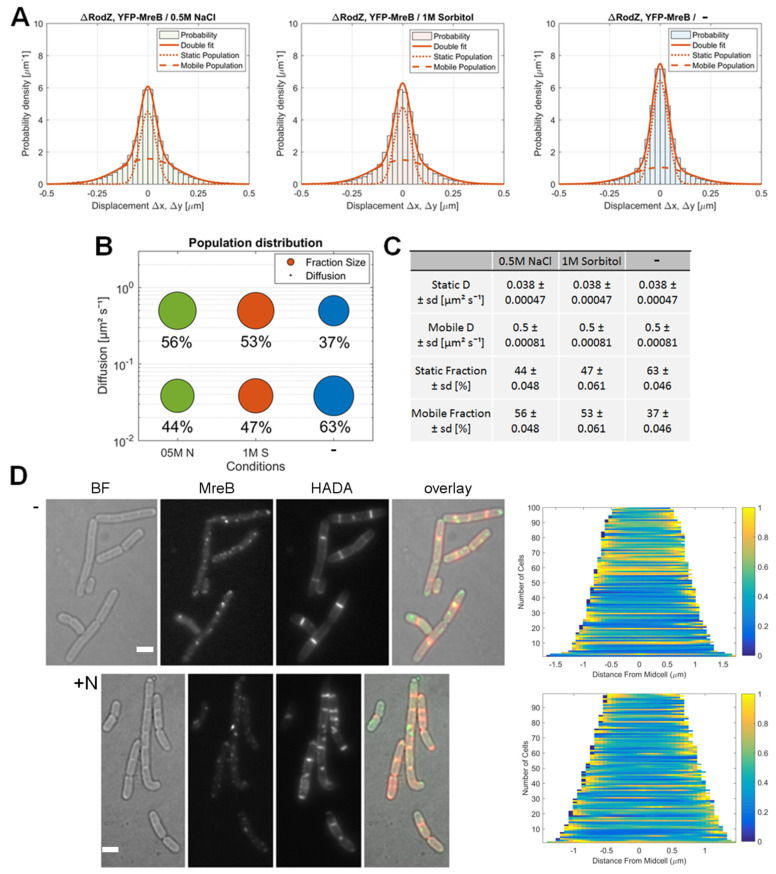
Adaption of MreB dynamics in response to osmotic stress in a *ΔrodZ* background. (**A**): Two-population Gaussian mixture model (GMM) fit of displacement vs. probability density for YFP-MreB under the control of the xylose promotor (+0.01% xylose) in a *ΔrodZ* background, in S7_50_ minimal media under normal growth conditions (blue circles) (-) and with the addition of 500 mM NaCl (green circle) (05M N) or 1 M sorbitol (red circles) (1M S); (**B**): Bubble plot of the diffusive populations (relative fraction sizes, D [μm^2^ s^–1^]) as identified by GMM curve fit in A; (**C**): Corresponding table of diffusion and relative fraction sizes for the slow-mobile and mobile populations; (**D**): Bright field (BF), YFP-MreB (green-channel), HADA (red-channel, 0.5 mM) and overlay images of *B. subtilis* cells (*ΔrodZ*) in the exponential phase, expressing YFP-MreB in S7_50_ media and the corresponding demograph of the distribution of HADA signal throughout *n* = 100 cells (20 min HADA staining without stress (-) and with added 0.5 M NaCl, images taken after washing 3 times with PBS), scale bar 2 μm.

**Figure 4 microorganisms-12-01309-f004:**
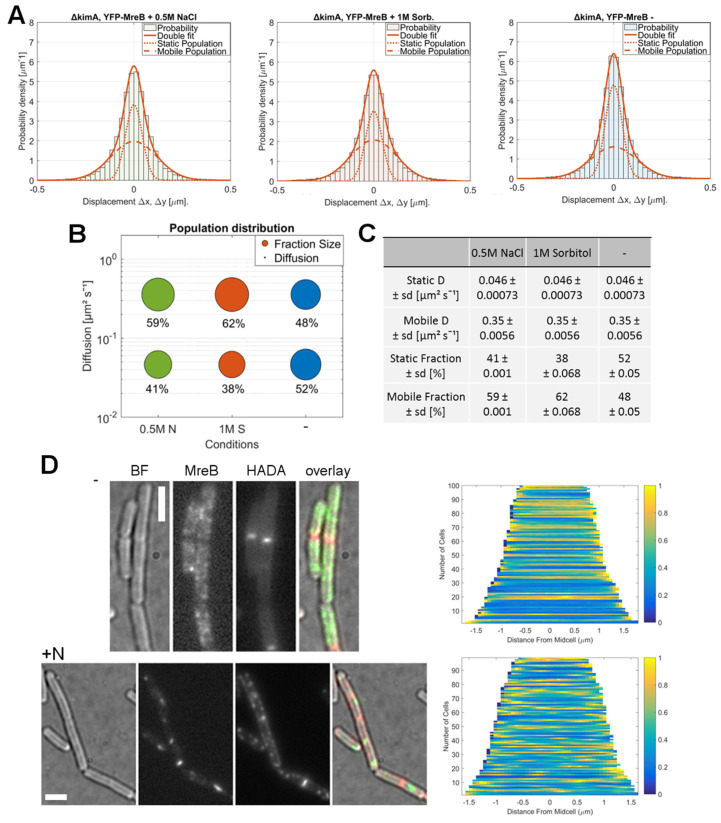
Adaption of MreB dynamics in response to osmotic stress in a *ΔkimA* background. (**A**): Two-population Gaussian mixture model (GMM) fit of displacement vs. probability density for YFP-MreB under the control of the xylose promotor (+0.01% xylose) in a *ΔkimA* background, in S7_50_ minimal media under normal growth conditions (-) and with the addition of 500 mM NaCl (0.5M N) or 1 M sorbitol (1M S); (**B**): Bubble plot of the diffusive populations (relative fraction sizes, D) as identified by GMM curve fit in A; (**C**): Corresponding table of diffusion constants and relative fraction sizes for the slow-mobile and mobile populations; (**D**): Bright field (BF), YFP-MreB (green-channel), HADA (red-channel, 0.5 mM) and overlay images (MreB: green, HADA: red) of *B. subtilis* cells (*ΔkimA*) in the exponential phase, expressing YFP-MreB in S7_50_ media and the corresponding demograph of the distribution of HADA signal throughout *n* = 100 cells (20 min HADA staining without stress (-) and with added 0.5 M NaCl, images taken after washing 3 times with PBS), scalebar 2 μm.

**Figure 5 microorganisms-12-01309-f005:**
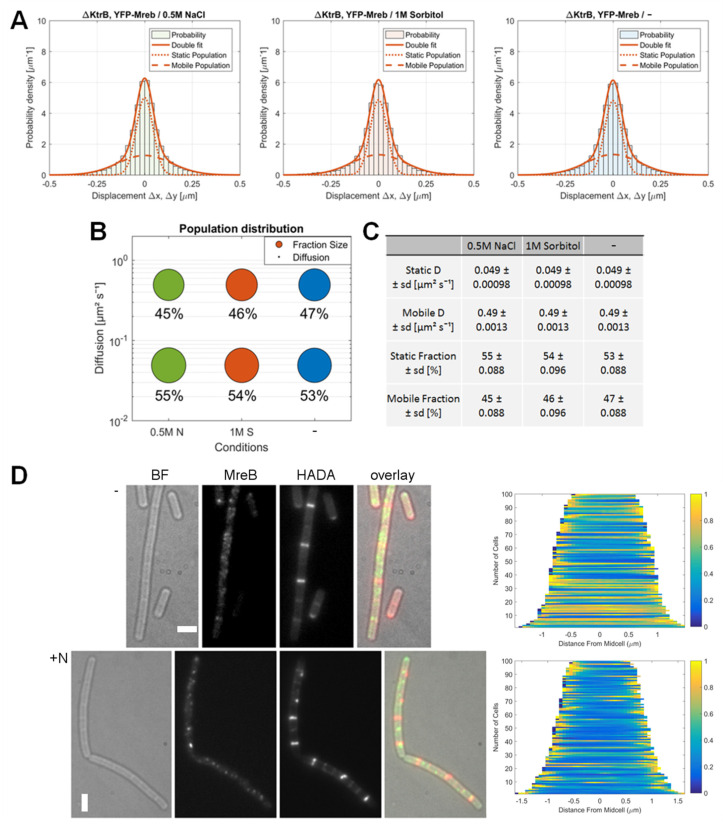
Adaption of MreB dynamics in response to osmotic stress in a *ΔktrB* background. (**A**): Two-population Gaussian mixture model (GMM) fit of displacement vs. probability density for YFP-MreB under the control of the xylose promotor (+0.01% xylose) in a *ΔktrB* background, in S7_50_ minimal media under normal growth conditions (-) and with the addition of 500 mM NaCl (0.5M N) or 1 M sorbitol (1M S); (**B**): Bubble plot of the diffusive populations (relative fraction sizes, D [μm^2^s^−1^]) as identified by GMM curve fit in A; (**C**): Corresponding table of diffusion and relative fraction sizes for the slow-mobile and mobile populations; (**D**): Bright field (BF), YFP-MreB (green-channel), HADA (red-channel, 0.5 mM) and overlay images (MreB: green, HADA: red) of *B. subtilis* cells (*ΔktrB*) in the exponential phase, expressing YFP-MreB under the control of the xylose promotor (+0.01% xylose) in S7_50_ media and the corresponding demograph of the distribution of HADA signal throughout *n* = 100 cells (20 min HADA staining without stress (-) and with added 0.5 M NaCl, images taken after washing 3 times with PBS), scale bar 2 μm.

**Figure 6 microorganisms-12-01309-f006:**
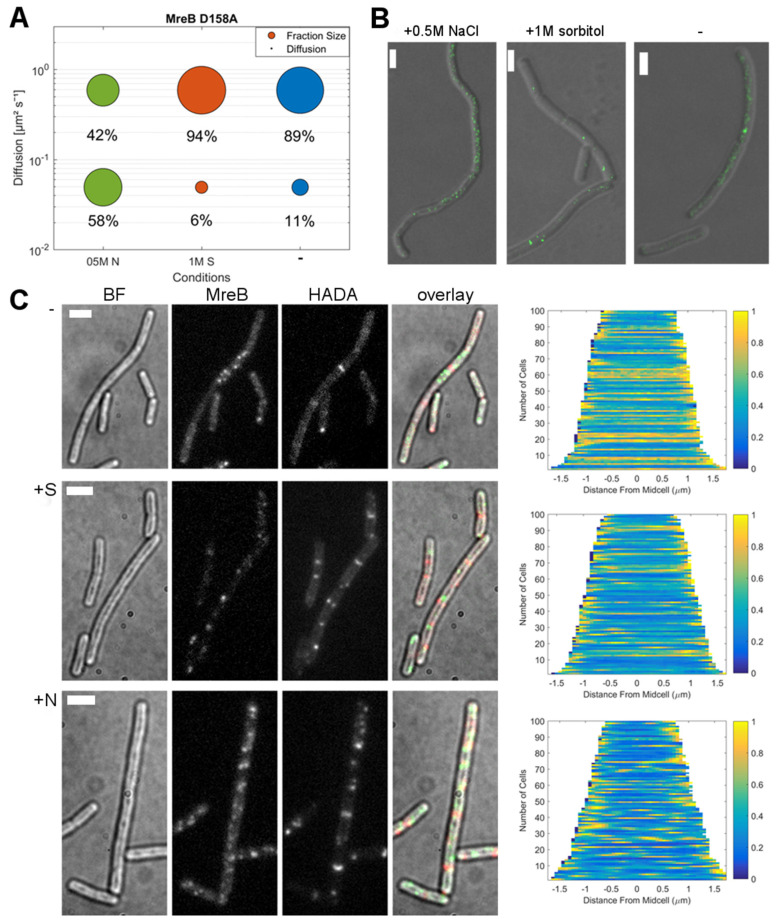
Adaption of MreB D158A dynamics in response to osmotic stress. (**A**): Bubble plot of the diffusive populations (relative fraction sizes, D. [μm^2^s^−1^]), from a two-population Gaussian mixture model (GMM) fit for YFP-MreB (D158A) under the control of the xylose promotor (+0.01% xylose) in S7_50_ minimal media under normal growth conditions (-) and with the addition of 500 mM NaCl (05M N) or 1 M sorbitol (1M S); (**B**): STED images of *B. subtilis* PY79 cells expressing YFP-MreB (D158A), under normal growth conditions (-) and with the addition of 500 mM NaCl (05M N) or 1 M sorbitol (1M S), scale bar 2 μm; (**C**): Bright field (BF), YFP-MreB (green-channel), HADA (red-channel, 0.5 mM) and overlay images (MreB: green, HADA: red) of *B. subtilis* cells (*ΔrodZ*) in the exponential phase, expressing YFP-MreB under the control of the xylose promotor (+0.01% xylose) in S7_50_ media and the corresponding demograph of the distribution of HADA signal throughout *n* = 100 cells (20 min HADA staining without stress (-) and with added 0.5 M NaCl, images taken after washing 3 times with PBS), scalebar 2 μm.

**Figure 7 microorganisms-12-01309-f007:**
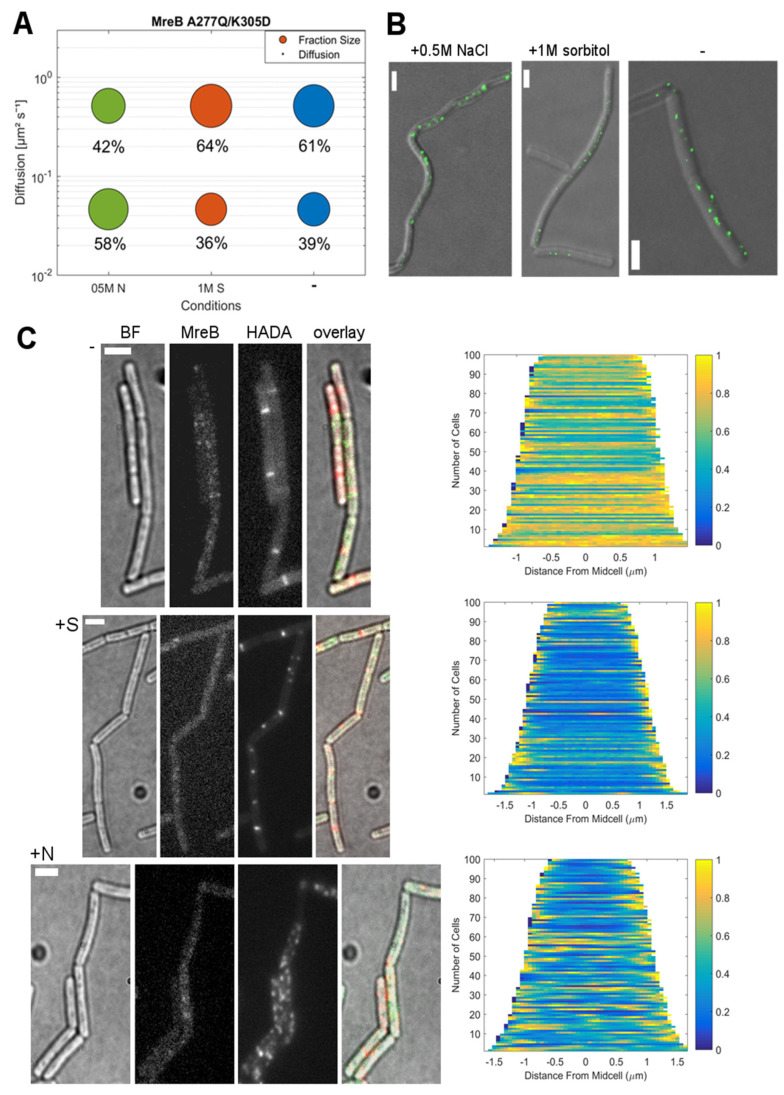
Adaption of MreB A277Q/K305D dynamics in response to osmotic stress. (**A**): Bubble plot of the diffusive populations (relative fraction sizes, D. [μm^2^s^−1^]), from a two-population Gaussian mixture model (GMM) fit for YFP-MreB (D158A) under the control of the xylose promotor (+0.01% xylose) in S7_50_ minimal media under normal growth conditions (-) and with the addition of 500 mM NaCl (05M N) or 1 M sorbitol (1M S); (**B**): STED images of *B. subtilis* PY79 cells expressing YFP-MreB (D158A), under normal growth conditions (-) and with the addition of 500 mM NaCl (05M N) or 1 M sorbitol (1M S), scale bar 2 μm; (**C**): Bright field (BF), YFP-MreB (green-channel), HADA (red-channel, 0.5 mM) and overlay images (MreB: green, HADA: red) of *B. subtilis* cells (*ΔrodZ*) in the exponential phase, expressing YFP-MreB under the control of the xylose promotor (+0.01% xylose) in S7_50_ media and the corresponding demograph of the distribution of HADA signal throughout *n* = 100 cells (20 min HADA staining without stress (-) and with added 0.5 M NaCl, images taken after washing 3 times with PBS), scalebar 2 μm.

## Data Availability

The original contributions presented in the study are included in the article/Appendix A, further inquiries can be directed to the corresponding author.

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
