# Peer review of "Adaptation of Bacillus subtilis MreB Filaments to Osmotic Stress Depends on Influx of Potassium Ions"

_microorganisms, 2024, doi:10.3390/microorganisms12071309_

Round 1

Reviewer 1 Report

Comments and Suggestions for Authors

Dear Editor,  

I would like to thank you for your confidence in reviewing this manuscript.

I send you here my comments for the manuscript review.

Title: Adaptation of MreB filaments to osmotic stress depends on influx of potassium ions.

The study was properly designed and performed and the manuscript has been written in an understandable manner. I recommend its publication after revision.

Major questions

Introduction

Line 56: “between regulation of cell with and enzymes”. This is uncompleted.

Line 8: Replace “I this work, we show” by “In this work, we showed”.

Results

Lines 113-120: Fig 1B citation is missed. Please, add it before Fig 1C.

Some figures like Fig 1 need to clear (high resolution), please, check and improve them.

References

Check all references form in the list and in the main text.

Author Response

Major questions

Introduction

Line 56: “between regulation of cell with and enzymes”. This is uncompleted.

- corrected “…or between the regulation of cell width and enzymes of the central carbon metabolism [13].”

Line 83: Replace “I this work, we show” by “In this work, we showed”.

- done

Results

Lines 113-120: Fig 1B citation is missed. Please, add it before Fig 1C.

- This was added in line 119: “The distribution of YFP-RodZ was visibly different from that of YFP-MreB (Fig. 1B, see further below).”

Some figures like Fig 1 need to clear (high resolution), please, check and improve them.

- This may be due to the pdf conversion; we will ensure high image quality during proofreading

References

Check all references form in the list and in the main text.

- we found a missing site for references in line 112 and corrected this. All other references were checked again.

Reviewer 2 Report

Comments and Suggestions for Authors

1. Since all the results were obtained for Bacillus subtilis, it is perhaps necessary to indicate this object of research in the title of the manuscript. 2. In the description of the construction of the strain with YFP-RodZ, it is written that it was obtained by a single cross-insertion of a plasmid. It is known that single-cross inserts are often deleted from the chromosome. Therefore, it would be good if the authors indicated how they controlled the integrity of this construct in the chromosome.

Author Response

  1. Since all the results were obtained for Bacillus subtilis, it is perhaps necessary to indicate this object of research in the title of the manuscript. 

- Title was changed to “Adaptation of Bacillus subtilis MreB filaments to osmotic stress depends on influx of potassium ions”

  1. In the description of the construction of the strain with YFP-RodZ, it is written that it was obtained by a single cross-insertion of a plasmid. It is known that single-cross inserts are often deleted from the chromosome. Therefore, it would be good if the authors indicated how they controlled the integrity of this construct in the chromosome.

- The entire plasmid can not excise from the chromosome because cells would lose the resistance cassette and die, but we have studied exponentially growing cells. We have added how the integrity of the construct was verified, in line 490: “The integrity of the strain was verified by a) sequencing of a PCR fragment generated by primers binding within pxyl (driving the expression of yfp-rodZ) and downstream of the 500 inserted rodZ bases (ensuring proper genome integration), and by Western blotting, showing a band at 52 kDa (ensuring that full length YFP-RodZ is produced).”